# Comparative analysis of dCas9-VP64 variants and multiplexed guide RNAs mediating CRISPR activation

**Kohei Omachi, Jeffrey H. Miner** [ORCID] *

Division of Nephrology, Washington University School of Medicine, St. Louis, Missouri, United States of America

* minerj@wustl.edu

## Abstract

CRISPR/Cas9-mediated transcriptional activation (CRISPRa) is a powerful tool for investigating complex biological phenomena. Although CRISPRa approaches based on the VP64 transcriptional activator have been widely studied in both cultured cells and in animal models and exhibit great versatility for various cell types and developmental stages *in vivo*, different dCas9-VP64 versions have not been rigorously compared. Here, we compared different dCas9-VP64 constructs in identical contexts, including the cell lines used and the transfection conditions, for their ability to activate endogenous and exogenous genes. Moreover, we investigated the optimal approach for VP64 addition to VP64- and p300-based constructs. We found that MS2-MCP-scaffolded VP64 enhanced basal dCas9-VP64 and dCas9-p300 activity better than did direct VP64 fusion to the N-terminus of dCas9. dCas9-VP64+MCP-VP64 and dCas9-p300+MCP-VP64 were superior to VP64-dCas9-VP64 for all target genes tested. Furthermore, multiplexing gRNA expression with dCas9-VP64+MCP-VP64 or dCas9-p300+MCP-VP64 significantly enhanced endogenous gene activation to a level comparable to CRISPRa-SAM with a single gRNA. Our findings demonstrate improvement of the dCas9-VP64 CRISPRa system and contribute to development of a versatile, efficient CRISPRa platform.

## Introduction

CRISPR (Clustered Regularly Interspaced Short Palindromic Repeats)/Cas9-mediated activation (CRISPRa)-based regulation of gene expression is a powerful tool for understanding complex biological phenomena, because it allows for the simultaneous regulation of multiple genes. CRISPRa has been used in broad fields of research, including direct cell reprogramming by controlling master transcription factors that regulate cell lineage [1–3], cancer modeling by activating oncogenes [4], and therapeutic approaches by activating disease-modifying genes [5] and genes deficient due to haploinsufficiency [6].

The first generation version of CRISPRa, dCas9-VP64, was developed by fusing VP64, a strong transcriptional activator, to dead Cas9 (dCas9) [7], which has no nuclease activity, allowing activation of guide RNA (gRNA) -targeted endogenous genes [8, 9]. Subsequently,

**Data Availability Statement:** All relevant data are within the paper and its Supporting Information files.

**Funding:** This work was supported by grants from the National Institutes of Health https://www.nih.

gov/ (R01DK058366, R01DK078314, and R01DK128660 to J.H.M.), the Children's Discovery Institute of Washington University and St. Louis Children's Hospital https://www.stlouischildrens.org/about-us/childrens-discovery-institute (to J.H.M.), the Japan Society for the Promotion of Science (JSPS https://www.jsps.go.jp/english/) Program for Postdoctoral Fellowships for Research Abroad (to K.O.) and the Cell Science Research Foundation (https://www.shionogi.com/jp/ja/sustainability/society/social-contribution-activities/foundation/zaidan.html) Program for Fellowships for Early Career Researcher (B12021-001 to K.O.). The funders had no role in study design, data collection and analysis, decision to publish, or preparation of the manuscript.

**Competing interests:** The authors have declared that no competing interests exist.

protein tagging systems called SunTag (dCas9-SunTag-VP64) [10] and MS2-MCP (dCas9-VP64 + MCP-VP64) [11] were developed to increase the number of VP64s at the same locus and enhance activation efficiency. In a similar approach, VP64-dCas9-VP64, in which one VP64 was added to the N-terminus of dCas9-VP64, exhibited increased efficiency of transcriptional activation [1]. More recently, additional transcriptional activators such as p65, Rta, and HSF1 have been identified, and CRISPRa-VPR [2], SAM [11], SPH [12] and TREE [13] systems have been developed by combining multiple transcriptional activators. Based on a different concept regarding transcriptional inducers, dCas9 fused to epigenetic modifiers such as p300, histone acetylase [14] and Tet1, a CpG DNA demethylase [15], have also been used to activate endogenous genes.

Although these studies have significantly contributed to the development of CRISPRa technology, most of them have been conducted in *in vitro* and *ex vivo* cell culture systems, but there are still challenges regarding *in vivo* applications. Specifically, several studies have reported on the *in vivo* toxicity of CRISPRa components. For example, it has been reported that VPR and SAM are toxic when highly expressed in *Drosophila* with a strong promoter [16]. In mice, ubiquitous expression of VPR during development and expression in inhibitory neurons are toxic [17]. On the other hand, ubiquitous expression of Cas9 [18] and dCas9-VP64 [6] in mice is not overtly toxic, suggesting that neither dCas9 nor VP64 is inherently toxic *in vivo*, and that high expression of either p65 or Rta, or both, may be responsible for the observed toxicity. Therefore, it is crucial to develop CRISPRa technologies that take into account both the efficiency of gene activation and the toxic side effects when considering their use for *in vivo* applications. Based on these findings, we hypothesized that a simple VP64-based CRISPRa would be useful *in vivo* in a variety of cell types, developmental stages, and pathological conditions. However, although there have been comparative studies of next-generation CRISPRa constructs such as SAM, VPR, and SPH with high transcriptional activation capabilities [12, 13, 19–24], to the best of our knowledge there are not enough studies that seek to improve and characterize VP64-based CRISPRa by directly comparing activities in a systematic, controlled fashion. Here, we aimed to characterize and rationally design approaches for VP64-based transcriptional activation of both endogenous and exogenous genes. We also investigated the effectiveness of multiplexed gRNAs at driving expression of individual genes.

## Methods

### Cell culture

HEK293T cells were maintained in Dulbecco's Modified Eagle Medium (DMEM) (Gibco, #11885092) supplemented with 10% heat inactivated fetal bovine serum (FBS) (Gibco, #26140079) and 1% penicillin/streptomycin at 37°C in a humidified 5% $CO_2$ incubator. Neuro-2a cells were maintained in Minimum Essential Media (MEM) (Gibco, #11095080) supplemented with 10% heat inactivated fetal bovine serum (FBS) (Gibco, #26140079) and 1% penicillin/streptomycin at 37°C in a humidified 5% $CO_2$ incubator. For transfecting cells, Lipofectamine 3000 (Invitrogen, #L3000015) was used by following the manufacturer's protocol. Briefly, 120,000–150,000 cells per 12-well plate were seeded. After 20–24 h, cells were transfected with 1 μg of total plasmid DNA, 2μL of p3000 reagent and 3 μL of Lipofectamine 3000 reagent in 5003B0043L of Opti-MEM (Gibco, #31985070). At 48 h post-transfection, transfected cells were used for q-RT-PCR analysis, fluorescence imaging and luciferase assays.

### Plasmid construction

lenti dCAS-VP64_Blast (Addgene plasmid #61425), lentiMPH v2 (Addgene plasmid #89308) and lenti sgRNA(MS2)_puro backbone (Addgene plasmid #73795) [11] were gifts from Feng

Zhang [11, 25]. Lenti_MCP-VP64_Hygro (Addgene plasmid #138458) was a gift from Jian Xu [26]. EF1α-dCas9-10xGCN4_Hygro and EF1α-scFv-p65-HSF1-Blast was generated by Gibson assembly cloning (NEB, #E5520S). Briefly, Lenti-dCas9-10xGCN4_Hygro and Lenti-scFv-p65-HSF_Blast were amplified from dSV40-NLS-dCas9-HA-NLS-NLS-10xGCN4 (Addgene plasmid #107310, a gift from Hui Yang) and EF1α-scFv-p65-HSF1-T2A-EGFP-WPRE-PolyA (Addgene plasmid #107311, a gift from Hui Yang) [12] and inserted into lentiMPH v2 and lenti dCAS-VP64_Blast backbone, respectively. Lenti-EF1α-dCas9-p300_Blast was generated by replacing the PuroR of pLV-dCas9-p300-P2A-PuroR (Addgene plasmid #83889, a gift from Charles Gersbach) [27] with BlastR using Gibson Assembly. Lenti-EF1α-VP64-dCas9-VP64_-Blast was generated by inserting VP64 in-frame right after the ATG start codon of lenti dCAS-VP64_Blast (Addgene plasmid #61425). Similarly, Lenti-EF1α-VP64-dCas9-p300_Blast was generated by inserting VP64 in-frame right after the ATG start codon of Lenti-EF1α-dCas9-p300_Blast. Lenti-EF1α-scFv-VP64_Blast was generated by fusing scFv-sfGFP-GB1 from pHRdSV40_scFv_GCN4_sfGFP_p65-hsf1_GB1_NLS (Addgene plasmid #79372, a gift from George Church) [19] and VP64 from lenti dCAS-VP64_Blast (Addgene plasmid #61425) using Gibson Assembly and inserting into the backbone of lenti dCAS-VP64_Blast (Addgene plasmid #61425). pCR-U6-gRNA-miniCMV-TdTomato was generated by inserting mini-CMV-TdTomato from reporter-gT1 (Addgene plasmid #47320, a gift from George Church) [28] into lenti sgRNA(MS2)_puro backbone (Addgene plasmid #73795); then U6-gRNA-mini-CMV-TdTomato was inserted into pCR Blunt II-TOPO. pCR-U6-gRNA-miniCMV-Nluc was generated by replacing TdTomato with Nluc from pNLF1-C [CMV/Hygro] (Promega #N1361) using Gibson Assembly. pCR-U6-gRNA-TRE3G-TdTomato and -Nluc were generated by replacing the miniCMV promoter of lenti sgRNA(MS2)_puro-miniCMV-TdTomato/Nluc with the TRE3G promoter from pTRE3G (Takara, #631173) using Gibson Assembly and inserting into pCR Blunt II-TOPO. The plasmids used in this study are listed in S1 Table. Newly generated plasmids from this study will be deposited into Addgene.

## gRNAs

The sequences of gRNAs used for activating endogenous and exogenous genes are listed in S1 Table. Oligonucleotides of 5'CACC-sense gRNA-3' and 5' AAAC-antisense gRNA-3' were purchased from Integrated DNA Technology (IDT). They were annealed by cooling from 95°C to 25°C for 1.5 hours. Annealing reaction mixtures were prepared as follows: 1μL of sense oligo (100μM), 1μL of antisense oligo (100μM), 10x annealing buffer (400 mM Tris-HCl (pH 8.0); 200 mM MgCl2; 500 mM NaCl) and 7 μL of nuclease free water. The annealed oligonucleotides were cloned into a gRNA expression vector by Golden Gate Assembly.

Golden Gate Assembly reactions were prepared as follows: 2.5 μL of annealed oligo, 1μL of gRNA cloning vector (80ng), 1μL of Esp3I restriction enzyme (NEB, #R0734S), 1.5μL of T4 DNA ligase (NEB, #M0202S), 2 μL 10x T4 DNA ligase buffer and 12.5 μL of nuclease free water. The Golden Gate Assembly reaction was performed in a thermal cycler using the following program: Step 1: 37°C for 5min; Step 2: 16°C for 5min; repeat steps 1–2 for 60 cycles; step 3: 75°C for 5min, step 4: 4°C hold. 5μL of reaction was transformed into NEB 5-alpha Competent E. coli (NEB, #C2987H) by following the manufacturer's protocol.

## RNA extraction and quantitative RT-PCR analysis

Cells were harvested 48 h post-transfection. Transfected cells were lysed with TRIzol reagent (Invitrogen, #15596018), and RNA was extracted and purified by following the manufacturer's

protocol. Purified RNA was quantified by A260/280 absorbance. cDNA was synthesized using the PrimeScript RT Master Mix (Takara, #RR036A) by following the manufacturer's protocol. Briefly, cDNA was synthesized using 62.5 ng of RNA per target gene at 37˚C for 30 min, and then RT enzyme was heat inactivated at 85˚C for 5 sec. For qPCR analysis, Fast SYBR Green Master Mix (Applied Biosystems, #4385612) was used by following the manufacturer's protocol. The sequences of primers used for qRT-PCR are listed in S1 Table.

### Luciferase assays

The protocol for dual luciferase assays was described previously [29]. Briefly, pCR4-U6-gRNA-miniCMV-NanoLuc or pCR4-U6-gRNA-TRE3G-NanoLuc and HSV-TK-Luc2 plasmids were transfected into HEK293T cells. At 48 h post transfection, transfected cells were washed once with phosphate buffered saline. Then, ONE-Glo Ex reagent was added, and firefly luciferase activity was measured. Subsequently, NanoDLR Stop & Go reagent (Promega, #N1620) was added, and the reaction plate was incubated for 10 min at room temperature, then NanoLuc luciferase activity was measured. The luciferase activity in the cell lysates was measured using a GloMax Navigator system (Promega). All luciferase assays were conducted in LumiNunc 96-well white plates (Thermo Scientific, #136101). NanoLuc luciferase activity was normalized to constitutively expressed firefly luciferase activity.

### Statistics

Statistical parameters are reported in the figure legends. Gene expression analysis was performed in triplicate using 3 independent transfections. Luciferase assays were performed in quadruplicate using 4 independent cell cultures. The significance of differences in multiple-groups was determined by analysis of variance (ANOVA) with Tukey-Kramer post-hoc. Differences with $P$ values of less than 0.05 were considered statistically significant. The results of a more detailed statistical analysis are described in the (S2 and S3 Tables).

## Results

### Comparative analysis of CRISPR activation platforms

To directly compare the ability of the various established CRISPRa systems to activate transcription of endogenous and exogenous genes, we generated expression plasmids for dCas9 and domains recruiting the VP64 transcriptional activator by SunTag and gRNA scaffolding using the same promoter and backbone. The CRISPRa-SPH [12]and SAM [11] systems were used as positive controls that promote high levels of transcriptional activation (Fig 1A). The degree of transcriptional activation in each system was assessed by quantifying increases in gene expression using previously validated gRNAs targeted to human *ASCL1* [11], *MYOD1* [11], and *NEUROD1* [19] and mouse *Neurog2* [11] and *Hbb-bh1* [19]. Consistent with previous reports, the CRISPRa-SPH and -SAM systems were superior to all the VP64-based CRISPRa systems that we tested, in both human and mouse cells (Fig 1B and 1C). This suggests that VP64-based CRISPRa has room for optimization and improvement.

### N-terminal VP64 addition to dCas9-VP64

Here we focused on the second generation VP64-dCas9-VP64 (2VP) [1], dCas9-VP64+ MCP-VP64 (VP+MV) [11], and the epigenetic regulator dCas9-p300 (p300) [14], which are more active than the first generation dCas9-VP64 (VP). Although dCas9-SunTag-VP64 [10] was also better than VP, it was not included in this study due to limitations in combining it with other systems. In agreement with previous reports, the present results (Fig 1B and 1C)

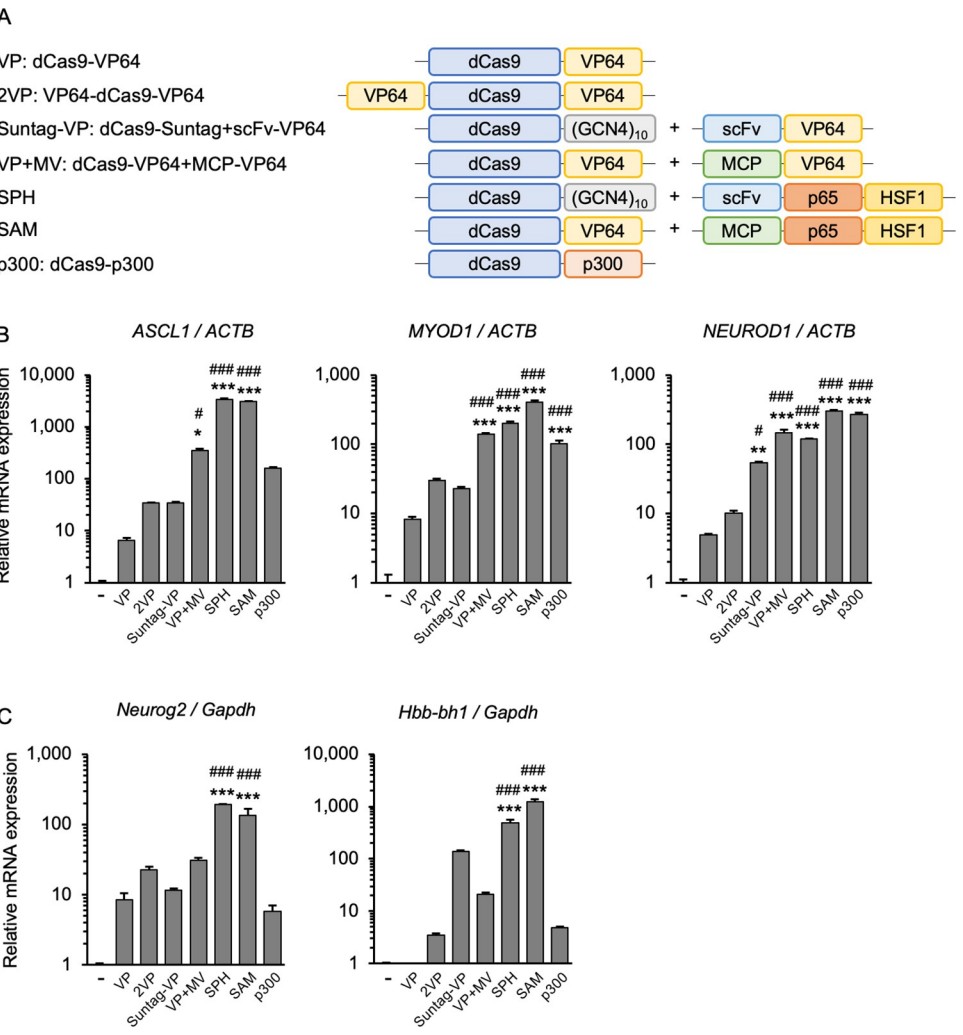

**Fig 1. Functional comparison of various CRISPRa approaches to activate endogenous genes in human and mouse cells.** (A) Schematic diagrams of the CRISPRa systems used in the comparative studies. (B) Expression analysis of three endogenous genes in human HEK293T cells. SAM and SPH induced the expression of *ASCL1* and *MYOD1* better than did the other systems. SunTag-VP, VP+MV and p300 similarly activated *ASCL1* and *MYOD1*, but p300 activated *NEUROD1* stronger than did the VP systems. (C) Expression analysis of two endogenous genes in mouse Neuro-2a cells. SAM and SPH induced both *Neurog2* and *Hbb-bh1* better than did the other systems. For activation of both human and mouse genes, increasing the number of VP64s, as in 2VP, Suntag-VP, and VP+MV, improved the efficiency compared with VP. Error bars indicate the mean ± SE (n = 3). Statistical analysis was performed using one-way ANOVA with Tukey's multiple comparisons test. **, $P < 0.01$; ***, $P < 0.005$ vs. non-induced control, #, $P < 0.05$; ###, $P < 0.005$ vs. VP.

showed that the addition of VP64 to the N-terminus of VP to make 2VP improves transcriptional induction. Therefore, in an attempt to improve VP+MV, we added VP64 to the N-terminus of dCas9 in VP+MV (Fig 2A, 2VP+MV). However, although 2VP+MV outperformed 2VP, it did not improve the ability of VP+MV to activate *ASCL1*, *MYOD1* and *NEUROD1* (Fig 2B).

## Combining the VP64 and dCas9-p300 systems

Next, we combined the p300 system with N-terminal VP64 and MCP-VP64 (Fig 3A). The combination of p300 with MV enhanced the transcriptional activation of *ASCL1* and

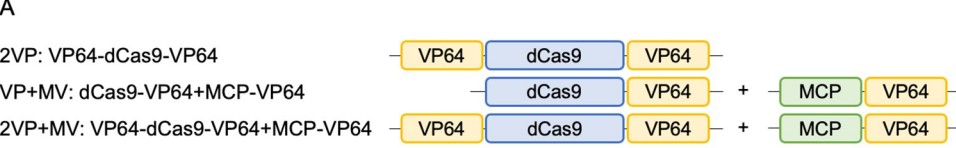

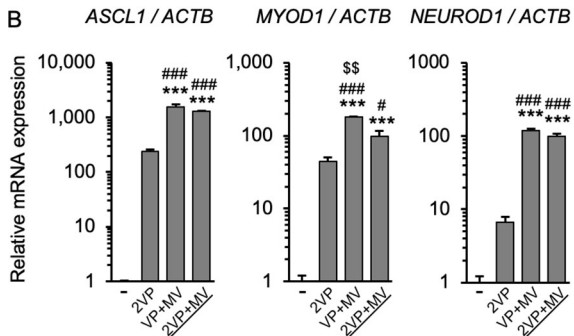

**Fig 2. Analysis of CRISPRa-VP64 with addition of N-terminal VP64.** (A) Schematic diagrams of the CRISPRa-VP64 systems tested; 2VP+MV is novel. (B) RNA expression of three endogenous genes was assayed in human HEK293T cells expressing the different CRISPRa systems. VP+MV showed the highest activity. Error bars indicate the mean ± SE (n = 3). Statistical analysis was performed using one-way ANOVA with Tukey's multiple comparisons test. ***, $P < 0.005$ vs. non-induced control, #, $P < 0.05$; ###, $P < 0.005$ vs. 2VP, $$, $P < 0.01$ vs. 2VP+MV.

*NEUROD1* by p300 alone. Transcriptional activation of *MYOD1* was equivalent in p300 and p300+MV. On the other hand, the addition of N-terminal VP64 decreased transcriptional induction for both p300 and p300+MV (Fig 3B).

## Multiplexing of gRNAs targeting single genes

Multiplexing of gRNAs targeting single genes has been shown to enhance transcriptional activation by dCas9-VP64 [8, 9] and other CRISPRa systems [30]. Here we focused on 2VP, VP+MV, and p300+MV (Fig 4A), which showed high activity with single gRNAs (Figs 2B and 3B), and compared them in the context of multiple gRNAs (Fig 4B). *ASCL1*, *MYOD1*, and *IL1RN* [14], for which several active gRNAs have already been identified, were used as representative genes. gRNA multiplexing enhanced *ASCL1*, *MYOD1*, and *IL1RN* expression compared with single gRNAs. Similar to the single gRNA studies, VP+MV and p300+MV showed higher activity than 2VP with multiplexed gRNAs (Fig 4C). Further analyses showed that the enhanced activity of VP+MV and p300+MV by gRNA multiplexing was better than that of SAM with single gRNA expression for activating *ASCL1* and *MYOD1*. In terms of *IL1RN* induction, the activity of VP-MV and p300+MV was improved by gRNA multiplexing, but SAM with single gRNA was still more active (Fig 4D).

## Activation of exogenous reporter genes

Finally, we utilized minimal CMV [2] and TRE3G [31] promoters, which have low basal activities, as targets for transcriptional activation of exogenous genes by CRISPRa. As shown in Fig 5B, we generated four reporter gene constructs: minimal CMV-TdTomato and -NanoLuc with one gRNA binding site; and TRE3G-TdTomato and -NanoLuc with seven identical gRNA binding sites. Therefore, the TRE3G reporters should exhibit higher sensitivity to activation than the CMV reporters. None of these constructs showed reporter protein expression at baseline, but addition of CRISPRa constructs (Fig 5A) effectively induced reporter protein

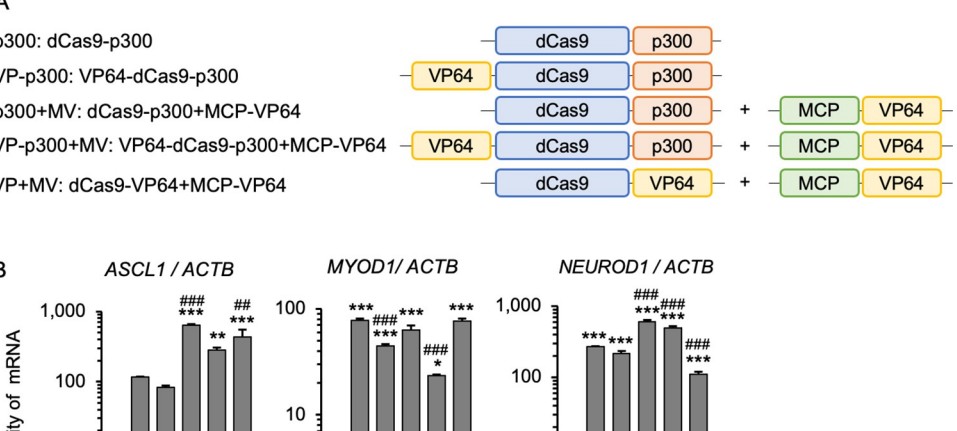

**Fig 3. Analysis of combined VP64 and dCas9-p300 CRISPRa systems.** (A) Schematic diagrams of the CRISPRa systems evaluated. VP-p300 and VP-p300+MV are novel. (B) RNA expression of three endogenous genes was tested in human HEK293T cells transfected with the indicated CRISPRa systems. The addition of VP64 to p300 by MS2-MCP scaffolding (to make p300+MV) enhanced transcriptional activation vs. p300 alone. Direct fusion of VP64 to the N-terminus of dCas9 (to make VP-p300) reduced the activity of transcriptional induction vs. p300 alone. Addition of VP64 to dCas9-p300 by MS2-MCP-VP64 (to make p300+MV) was more advantageous than direct fusion of VP64 (to make VP-p300+MV and VP-p300+MV). Error bars indicate the mean ± SE (n = 3). Statistical analysis was performed using one-way ANOVA with Tukey's multiple comparisons test. *, $P < 0.05$; **, $P < 0.01$; ***, $P < 0.005$ vs. non-induced control, ##, $P < 0.01$; ###, $P < 0.005$ vs. p300.

expression (Fig 5C and 5D). Similar to the activation of endogenous genes, VP+MV and p300 +MV significantly induced both tdTomato and NanoLuc expression (Fig 5C and 5D). For the TRE3G reporter, with seven gRNA binding sites, the enhancement of reporter protein expression by p300+MV was limited compared to VP+MV. The spatial constraints of p300, a relatively large protein, could be responsible for this surprising finding.

## Discussion

CRISPR/Cas9-mediated transcriptional regulation has been used in broad research fields, including direct cell reprogramming [1–3], disease modeling [4], and therapeutic applications [5, 6]. Activation of endogenous genes by CRISPRa is a powerful tool for investigating cell biological processes requiring complex genetic regulation because of its ability to control multiple genes simultaneously. Moreover, the size of the target gene is not a limiting factor. Such technologies are important for hypothesis testing in the omics era [32–35].

In addition to comparative studies of existing systems, we rationally designed new versions of CRISPRa-VP64 with different modes of VP64 recruitment. Our strategies were as follows: 1) addition of VP64 to the N-terminus of dCas9 in dCas9-VP64+MCP-VP64; and 2) combination of VP64 and dCas9-p300 systems by either direct fusion or MS2-MCP scaffold. The addition of VP64 to the N-terminus of dCas9-VP64 had already been shown to increase the efficiency of transcriptional activation [1], but it was unclear whether the same effect would be observed for dCas9-VP64+MCP-VP64. Although the combination of MCP-VP64 with dCas9-p300 increases transcriptional activation when targeted to enhancers [26], transcriptional activation when targeted to promoter regions has not been well studied. In addition,

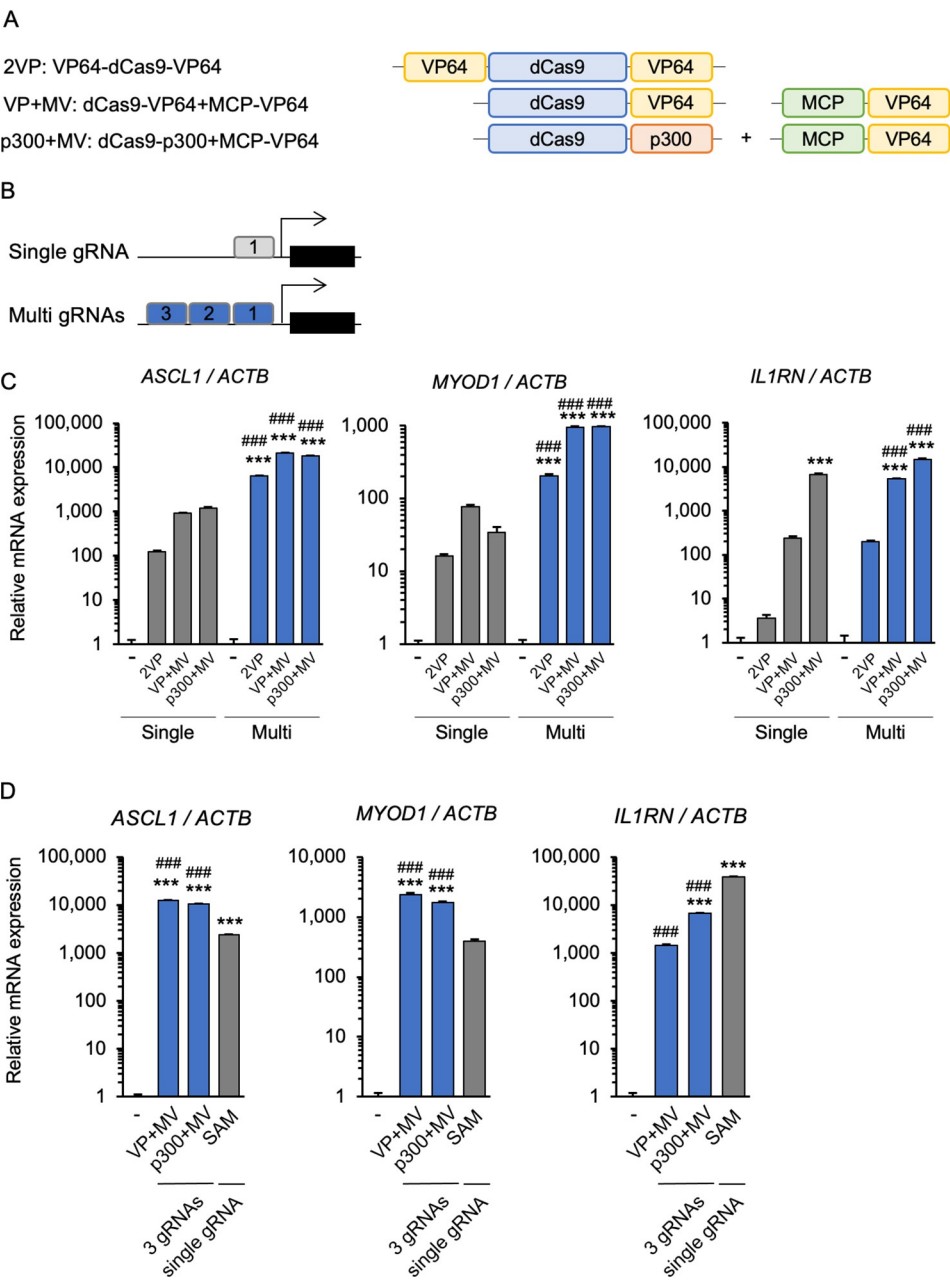

**Fig 4. Analysis of the impact of multiplexing gRNAs targeting a single gene.** (A) Schematic diagrams of CRISPRa systems used for gRNA multiplexing to target single genes. (B) Schematic diagrams show the positions of single and multiple gRNAs near the transcriptional start site. (C) Expression of three endogenous genes in the presence of a single gRNA or multiple gRNAs in human HEK293T cells. Multiplexed gRNAs enhanced transcription in all three systems tested. Error bars indicate the mean ± SE (n = 3). Statistical analysis was performed using one-way ANOVA with Tukey's multiple comparisons test. ***, $P < 0.005$ vs. non-induced control, ###, $P < 0.005$ vs. the level observed for each corresponding single gRNA experiment. (D) Expression of three endogenous genes were tested in CRISPRa-transfected human HEK293T cells. VP+MV and p300+MV with multiple gRNAs showed higher *ASCL1* and *MYOD1* activation than SAM with a single gRNA. For induction of *IL1RN*, VP+MV and p300+MV with multiple gRNAs did not reach the activity of SAM with a single gRNA. Statistical analysis was performed using one-way ANOVA with Tukey's multiple comparisons test. ***, $P < 0.005$ vs. non-induced control, ###, $P < 0.005$ vs. SAM.

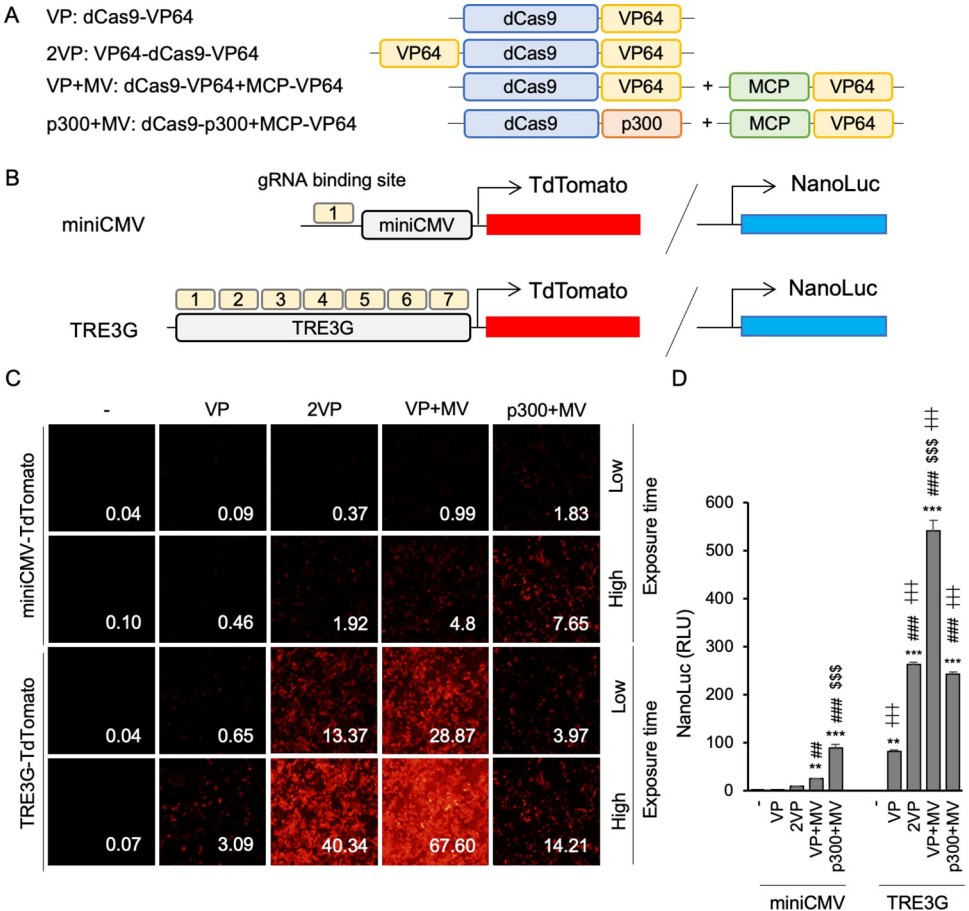

**Fig 5. CRISPRa-mediated transcriptional activation of exogenous TdTomato and NanoLuc reporter constructs.**
(A) Schematic diagrams of the CRISPRa-VP64 and -p300 systems used. (B) Schematic diagrams of TdTomato and NanoLuc reporter gene constructs. The miniCMV-driven reporter construct has one gRNA binding site, whereas the TRE3G-driven reporter construct has seven identical gRNA binding sites. (C) TdTomato fluorescence in HEK293T cells transfected with the indicated reporter and CRISPRa constructs. TdTomato expression was higher for TRE3G vs. miniCMV. The strength of TRE3G induction was in the order of VP+MV>2VP>p300+MV>VP. Fluorescence intensity is shown in the lower right corner of each image. (D) NanoLuc luminescence was measured in HEK293T cells transfected with the indicated reporter and CRISPRa constructs. The relative luminescence was higher for TRE3G vs. miniCMV. The strength of TRE3G induction was in the order of VP+MV>2VP>p300+MV>VP. Error bars indicate the mean ± SE (n = 4). Statistical analysis was performed using one-way ANOVA with Tukey's multiple comparisons test. **, $P < 0.01$, ***, $P < 0.005$ vs. non-induced control, ##, $P < 0.01$, ###, $P < 0.005$ vs. VP, $$$, $P < 0.005$ vs. 2VP, , $P < 0.005$ vs. each corresponding miniCMV-driven level of luminescence.

there were no studies on whether the addition of VP64 to the N-terminus of dCas9-p300 would enhance activity as it does for dCas9-VP64.

Our attempts to further improve dCas9-VP64+MCP-VP64 and dCas9-p300+MCP-VP64 by adding N-terminal VP64 were unsuccessful. Direct fusion of VP64 to the N-terminus of dCas9 was effective for dCas9-VP64, which has only a single VP64, but our results suggest that adding one VP64 is not effective for dCas9-VP64+MCP-VP64, which already has an amplified number of VP64s via the MS2-MCP scaffold. Contrary to our expectations, fusion of VP64 to the N-terminus of dCas9-p300 resulted in decreased activity compared to that observed with dCas9-p300 alone. Considering that the addition of VP64 by MV on p300 additively enhanced transcriptional induction of *ASCL1* and *NEUROD1*, N-terminal VP64 addition may decrease the ability of the dCas9 complex to access the target site, either by increasing its size or

changing protein conformation. The importance of such spatial accessibility has been reported regarding the incorporation of the DNA demethylase Tet1-CD for CRISPRa using the SunTag system [36]. Since the p300 core domain is a relatively large protein domain, like Tet1-CD, spatial accessibility is likely to be involved.

Our results also demonstrated that when targeting a region of a single gene with multiple gRNAs, it is necessary to consider the spatial accessibility of the gRNAs due to the spacing of their target sequences. We were able to evaluate the enhancement of activity by gRNA multiplexing in exogenous genes by utilizing the miniCMV promoter reporter, which has only one gRNA binding site, and the TRE3G promoter reporter, which has seven gRNA binding sites (Fig 5). dCas9-VP64, VP64-dCas9-VP64, and dCas9-VP64+MCP-VP64 showed increased activity with increased target sites in the TRE3G promoter. On the other hand, dCas9-p300+-MCP-VP64, which has the largest size, showed only limited increases in activity with increased numbers of binding sites in the TRE3G promoter compared to other systems.

Multiplexing gRNAs targeting a single gene enhances transcriptional activation in various CRISPRa platforms. Chavez et al. demonstrated that combining multiple gRNAs enhanced target gene expression in the dCas9-VP64, VPR, SAM and SunTag systems [19], but other dCas9-VP64 systems were not tested. In the present study we carefully and methodically compared VP64-dCas9-VP64, dCas9-VP64+MCP-VP64 and dCas9-p300+MCP-VP64 systems in the presence of both a single gRNA and multiplexed gRNAs targeting single endogenous genes and exogenous genes. The results indicated that dCas9-VP64+MCP-VP64 and dCas9-p300+-MCP-VP64 exhibit higher and more stable activity than VP64-dCas9-VP64, whether single or multiple gRNAs were used (Fig 4). However, as mentioned above, dCas9-p300+MCP-VP64 has spatial limitations, and the size of the coding sequence is larger than that of other CRISPRa constructs. If enhancer control is not the goal, VP64-dCas9-VP64 and dCas9-VP64+-MCP-VP64 are easier to handle for packaging into a single lentivirus and for assembling a transgenic construct.

## Conclusion

Based on our comparative analyses, dCas9-VP64 and -p300 systems were significantly improved by combining with MCP-VP64, but not by direct fusion of VP64 to the N-terminus of dCas9. Moreover, we showed that gRNA multiplexing further enhanced target gene expression in dCas9-VP64+MCP-VP64 and dCas9-p300+MCP-VP64. Thus, our findings support the improvement of the dCas9-VP64 system and contribute to developing a versatile and efficient CRISPRa platform.

## Supporting information

**S1 Table. Key resources (plasmids, gRNAs and primers).**
(DOCX)

**S2 Table. Statistical analysis (one-way ANOVA with Tukey-Kramer's or Dunnett's test).**
(XLSX)

**S3 Table. Statistical analysis (student's t-test).**
(XLSX)

## Author Contributions

**Conceptualization:** Kohei Omachi.

**Formal analysis:** Kohei Omachi.

**Funding acquisition:** Jeffrey H. Miner.

**Investigation:** Kohei Omachi.

**Methodology:** Kohei Omachi.

**Project administration:** Jeffrey H. Miner.

**Supervision:** Jeffrey H. Miner.

**Validation:** Kohei Omachi.

**Writing – original draft:** Kohei Omachi.

**Writing – review & editing:** Kohei Omachi, Jeffrey H. Miner.

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
