## [Decision Letter · Decision Letter 0]

14 Apr 2022

PONE-D-22-06035Comparative analysis of dCas9-VP64 variants and multiplexed guide RNAs mediating CRISPR activationPLOS ONE

Dear Dr. Miner,

Thank you for submitting your manuscript to PLOS ONE. The manuscript is reviewed by two independent reviewers, and both found it to be useful to the community and have minor concerns. Therefore, we invite you to submit a revised version of the manuscript that addresses the points raised during the review process.

We look forward to receiving your revised manuscript.

Kind regards,

Gaurav Varshney, Ph.D.

Academic Editor

PLOS ONE

Journal Requirements:

Reviewers' comments:

Reviewer's Responses to Questions

**Comments to the Author**

1. Is the manuscript technically sound, and do the data support the conclusions?

Reviewer #1: Yes

Reviewer #2: Yes

2. Has the statistical analysis been performed appropriately and rigorously? 

Reviewer #1: No

Reviewer #2: Yes

3. Have the authors made all data underlying the findings in their manuscript fully available?

Reviewer #1: Yes

Reviewer #2: Yes

4. Is the manuscript presented in an intelligible fashion and written in standard English?

Reviewer #1: Yes

Reviewer #2: Yes

5. Review Comments to the Author

Reviewer #1: In this study, authors analyzed multiple combinations of CRISPR/Cas9-mediated activation system. They found dCas9-VP64 and dCas9-p300 combined with MCP-VP64 is more efficient than direct fusion of VP64 to the N-terminus of dCas9 in transcriptional activation for both endogenous and exogenous genes. This study is well-defined and can provide researchers with another efficient tool for gene functional study. However, there are some points that should be revised and addressed by authors so that readers can better understand.

The major point is that the statistical comparisons are unclear and require better clarification.

- Figure 1: The statistical significance was labeled only on some bars. Does this mean only marked bars are significant compared to control? And please clarify the comparisons by drawing lines on the graphs for those were described in text or apply a different symbol as author showed in later figures. Why use two-way ANOVA in this figure?

- Figure 2: 2VP is not significant?

- Figure 3: Not significant for p300 and VP-p300 in ASCL1? p300+MV is not novel? Please double check the statistical significance of VP-p300+MV in ASCL1 and MYOD1.

- Figure4: Only single p300+MV of IL1RN is statistically significant when compared to control? In IL1RN, is multiplexing gRNAs compared to single gRNA of p300+MV that significant (***)?

- Figure 5: Check labeling for statistical comparisons. I see no description for the activity of VP+MV and p300+MV in IL1RN.

Minor points:

- Please unify the labeling by using either control or non-induced control (used in Figure 2) in figure legends.

- Author used abbreviation for all figures except Figure 6, please unify.

- Please consider combining similar figures, for example, Fig. 2 & 3, Fig. 4 & 5 and Fig. 6 & 7.

- Please consider using fluorescent intensity to convey the strength of reporter activation in Figure 6C.

Reviewer #2: The manuscript is a logical and useful extension of previous work on modified CRISPR tools. The methods used are standard and the outcomes are helpful to the research community. While not surprising, the experiment that shows the impact of multiplexing gRNAs targeting a single gene is definitely a useful to researchers.

---

## [Author Response · Author response to Decision Letter 0]

6 May 2022

Editor’s comments:

Thank you for submitting your manuscript to PLOS ONE. The manuscript is reviewed by two independent reviewers, and both found it to be useful to the community and have minor concerns. Therefore, we invite you to submit a revised version of the manuscript that addresses the points raised during the review process.

Reviewers' comments: 

Reviewer #1: In this study, authors analyzed multiple combinations of CRISPR/Cas9-mediated activation system. They found dCas9-VP64 and dCas9-p300 combined with MCP-VP64 is more efficient than direct fusion of VP64 to the N-terminus of dCas9 in transcriptional activation for both endogenous and exogenous genes. This study is well-defined and can provide researchers with another efficient tool for gene functional study. However, there are some points that should be revised and addressed by authors so that readers can better understand.

The major point is that the statistical comparisons are unclear and require better clarification.

- Figure 1: The statistical significance was labeled only on some bars. Does this mean only marked bars are significant compared to control? And please clarify the comparisons by drawing lines on the graphs for those were described in text or apply a different symbol as author showed in later figures. Why use two-way ANOVA in this figure?

RESPONSE: Thank you for taking a careful look at the data. Your thoughts are well taken. Even the groups that do not show a statistical significance are clearly different compared to the non-induced control. The differences are statistically significant with Student’s t test. However, all figures have multiple groups and we used one-way ANOVA multiple comparison analysis. Because there are groups that are clearly different but not significantly different by one-way ANOVA, we have added details of the TTEST and one-way ANOVA analyses to the supporting information (S2 and S3 Tables). Also, we have added a different symbol as we show in the later figures. We apologize for the error in our description. It is actually a one-way ANOVA, not a two-way ANOVA. 

- Figure 2: 2VP is not significant?

RESPONSE: 2VP is not statistically significant with one-way ANOVA. (But 2VP is statistically significant with Student’s t-test.). 

- Figure 3: Not significant for p300 and VP-p300 in ASCL1? p300+MV is not novel? Please double check the statistical significance of VP-p300+MV in ASCL1 and MYOD1.

RESPONSE: p300 and VP-p300 are not statistically significant with one-way ANOVA. (But they are statistically significant with Student’s t-test.)

- Figure4: Only single p300+MV of IL1RN is statistically significant when compared to control? In IL1RN, is multiplexing gRNAs compared to single gRNA of p300+MV that significant (***)?

RESPONSE: Only p300+MV is statistically significant with one-way ANOVA in the single gRNA expression. (All are statistically significant with Student’s t-test.). Comparison between multiplexing gRNAs to single gRNA of p300+MV is significant with one-way ANOVA (P <0.005).

- Figure 5: Check labeling for statistical comparisons. I see no description for the activity of VP+MV and p300+MV in IL1RN. 

RESPONSE: We have added description for the activity of VP+MV and p300+MV on IL1RN in the main text and figure legend. Since we combined Fig 4&5, they correspond to the current Fig 4D.

Minor points:

- Please unify the labeling by using either control or non-induced control (used in Figure 2) in figure legends.

RESPONSE: We apologize for the inconsistency of labels. We have unified the labeling by using non-induced control.

- Author used abbreviation for all figures except Figure 6, please unify.

- Please consider combining similar figures, for example, Fig. 2 & 3, Fig. 4 & 5 and Fig. 6 & 7.

RESPONSE: Thank you for suggestion. We have unified the abbreviation in all figures. Also, we have combined Fig. 4 & 5 and Fig. 6 & 7. 

- Please consider using fluorescent intensity to convey the strength of reporter activation in Figure 6C.

RESPONSE: Thanks for the idea to make the data more understandable to the reader. We have added the fluorescent intensity in the lower right corner of each image. 

Reviewer #2: The manuscript is a logical and useful extension of previous work on modified CRISPR tools. The methods used are standard and the outcomes are helpful to the research community. While not surprising, the experiment that shows the impact of multiplexing gRNAs targeting a single gene is definitely a useful to researchers.

RESPONSE: Thank you for your supportive comments.

---

## [Decision Letter · Decision Letter 1]

2 Jun 2022

Comparative analysis of dCas9-VP64 variants and multiplexed guide RNAs mediating CRISPR activation

PONE-D-22-06035R1

Dear Dr. Miner,

We’re pleased to inform you that your manuscript has been judged scientifically suitable for publication and will be formally accepted for publication once it meets all outstanding technical requirements.

Kind regards,

Gaurav Varshney, Ph.D.

Academic Editor

PLOS ONE

Additional Editor Comments (optional):

Reviewers' comments:

Reviewer's Responses to Questions

**Comments to the Author**

1. If the authors have adequately addressed your comments raised in a previous round of review and you feel that this manuscript is now acceptable for publication, you may indicate that here to bypass the “Comments to the Author” section, enter your conflict of interest statement in the “Confidential to Editor” section, and submit your "Accept" recommendation.

Reviewer #1: All comments have been addressed

2. Is the manuscript technically sound, and do the data support the conclusions?

Reviewer #1: Yes

3. Has the statistical analysis been performed appropriately and rigorously? 

Reviewer #1: Yes

4. Have the authors made all data underlying the findings in their manuscript fully available?

Reviewer #1: Yes

5. Is the manuscript presented in an intelligible fashion and written in standard English?

Reviewer #1: Yes

6. Review Comments to the Author

Reviewer #1: (No Response)

7. PLOS authors have the option to publish the peer review history of their article (what does this mean?). If published, this will include your full peer review and any attached files.

Reviewer #1: No

---

## [Editor Report · Acceptance letter]

8 Jun 2022

PONE-D-22-06035R1 

Comparative analysis of dCas9-VP64 variants and multiplexed guide RNAs mediating CRISPR activation 

Dear Dr. Miner:

I'm pleased to inform you that your manuscript has been deemed suitable for publication in PLOS ONE. Congratulations! Your manuscript is now with our production department. 

Kind regards, 

on behalf of

Dr. Gaurav Varshney 

Academic Editor

PLOS ONE